# Applying dyadic digital psychological interventions for reducing caregiver burden in the illness context: a systematic review and a meta-analysis protocol

Michelle Semonella ®,[1] Vanessa Bertuzzi ®,[2] Rachel Dekel,[3] Gerhard Andersson,[4,5] Giada Pietrabissa,[2,6] Noa Vilchinsky ®[1]

For numbered affiliations see end of article.

**Correspondence to**
Michelle Semonella;
michelle.semonella@biu.ac.il

## ABSTRACT

**Introduction** Providing informal care to one's romantic partner who is ill may become a highly distressing and demanding task. Based on the innovative dyadic coping model, several support interventions have been developed to alleviate informal caregivers' burden, including both caregivers' and care receivers' needs. Considering the unique challenges characterising the caregiving phenomenon, such as geographical barriers and time restrictions, digital solutions should be considered. However, there is a lack of research examining the effectiveness of dyadic digital solutions. Thus, this review aims to examine the existing literature on the efficacy of dyadic digital psychological interventions designed for caregivers and their care-receivers couples within the illness context.

**Methods and analysis** Randomised controlled trials targeting caregivers' burden among dyads of informal caregivers and care receivers will be identified via an electronic search of the following databases: PubMed, Embase, the Cochrane Library, Cinhal, Scopus, PsycINFO, MEDLINE and supplemented by hand searching of previous systematic reviews. The search will be undertaken following the PICO (population, intervention, comparison and outcome) elements. If possible, a meta-analysis will be conducted to examine: (1) the effectiveness of dyadic digital psychological interventions for reducing caregivers' burden (primary outcome) among caregivers who are in a romantic relationship with the care receivers; (2) the effectiveness of dyadic digital psychological interventions on secondary outcomes such as anxiety, depression, stress, quality of life, well-being and self-efficacy among caregivers and care receivers; and (3) moderating effects of clinical and methodological factors on caregivers' burden. Prior to inclusion in the review, retrieved papers will be critically appraised by two independent reviewers. The Cochrane Risk of Bias tool will assess the risk of bias for randomised controlled trials.

**Ethics and dissemination** Ethical approval is not required as no primary data will be collected. Findings will be disseminated through peer-reviewed publications, presentations at academic conferences and lay summaries for various stakeholders.

## STRENGTHS AND LIMITATIONS OF THIS STUDY

⇒ To increase the quality of included studies and reduce methodological heterogeneity, only studies assessed as having a low or moderate risk of bias, according to the Cochrane Collaboration's Risk of Bias tool 2.0, will be included.

⇒ Following the Centre and Reviews and Dissemination guidance, as well as the Preferred Items for Reporting of Systematic Reviews and Meta-Analyses Protocol guidelines, this protocol adopts independent study selection, independent data extraction and independent risk of bias assessment by two reviewers.

⇒ Only randomised controlled trials will be included to maintain a high-methodological standard.

⇒ The studies may not have collected all the variables of interest. Thus, we might not be able to examine all candidate moderators.

⇒ The review will be restricted to studies published in English, which might be indicative of a language bias.

**PROSPERO registration number** CRD42022299125.

# INTRODUCTION
## Informal caregiving

Informal care is defined as the unpaid care provided to an older, frail or ill person, by a person such as a spouse, parent, child, another relative, neighbour, friend or other non-kin.[1] The phenomenon of providing informal care is growing, mostly due to the growth in the ageing population, the increasing prevalence of chronic illnesses and more effective medical treatments saving lives. To date, it is estimated that 44 million people in Europe provide informal care to family or friends.[2] Indeed, in Europe, 80% of long-term care for non-independent people is provided by informal caregivers; this estimation makes them an essential element of

the care system in Europe.[3] At the same time, the growing number of caregivers has made caregiving a public health issue.[4 5] Being an informal caregiver can be a stressful experience and is often associated with negative psychosocial and health consequences as well as financial strains, all of which could increase caregiving burden.[6 7] Caregiver burden is a multidimensional concept consisting of physical, psychological, emotional, social and financial stressors linked with the caregiving experience which might contribute to caregivers' physical and psychological illnesses.[8–10] Indeed, due to their extensive caregiving responsibilities, many caregivers report physical exhaustion and even negative health consequences such as increased cardiovascular reactivity and poor immune system response.[11 12] It is therefore necessary to find ways to alleviate caregivers' burden and distress by developing suitable and feasible psychological support interventions or implementing existing ones. When designing or implementing interventions, it is important to take into account the characteristics, limitations and features that characterises informal care, such as its dyadic nature and caregivers' difficulties in terms of lack of time and money.

## Novel approaches to interventions for caregivers: the call for dyadic and digital solutions

During the last 20 years, there has been a shift in perspective in the health psychology literature: from an individual perspective of coping with the illness to a dyadic one.[13–15] Researchers have demonstrated that both members of the dyad actively share resources and manage stressors together as a couple. Specifically, a new conceptualization has emerged viewing illness as a 'we-disease',[16] which taps mutual influences between two coping partners in a dyad.[17–19] Particularly, providing care in romantic relationships differs from other kinds of relationships for three main reasons. First, romantic partners are most likely to live together with the care recipients and therefore usually provide support when needed. Second, caregivers who are in a romantic relationship are usually older than adult children who provide care for their parents; thus, they may have their own health problems which may lead to greater perceived stress while providing informal care. Third, the romantic partner is usually the most important attachment figure for the care receiver,[20] and many expectations are directed towards him/her on behalf of the patient.[21 22] Thus, caregiving spouses in particular might be at higher risk for caregiver burden.[23] Thus, given that caregiving is conceptualised as a dyadic process rather than an individualised one,[13] any intervention developed for caregivers should be designed dyadically, taking into account both caregivers' and care receivers' needs and consequences, as well as the relationship between the two. Indeed, different dyadic interventions have been developed to ease caregivers' stress and help them better cope with their condition in various health contexts, and they are effective in improving outcomes (such as caregiver burden, quality of life, stress,

anxiety and depression) in both informal caregivers and care receivers.[24–27]

Although dyadic interventions were shown to be successful and beneficial for both members of the dyad,[28–31] most of them have been delivered in person, requiring the participation of both members, which may limit their ultimate scalability and accessibility.[32] It is, therefore, necessary to develop interventions that are dyadic in nature—involving both partners—while at the same time being feasible. In recent years, using information technology has become a regular part of our daily life and also common in the clinical field and healthcare. During the COVID-19 pandemic, psychological support delivered through digital solutions became more relevant, accepted and feasible.[33] Indeed, the COVID-19 outbreak facilitated the acceptability and willingness to use information technology tools to deliver psychological interventions.[34] Thanks to psychologically based digital interventions, it is possible to reach patients from a distance, and research suggests that this mode of delivery can be as effective as other treatment formats[35] and serve as a complement or even alternative to regular face-to-face treatments.[36] Digital interventions would be particularly useful and efficient to fill gaps due to caregiving conditions, as they suit caregivers' needs in terms of lack of time, money and distance.[34] Moreover, digital solutions can have beneficial effects on caregivers' mental health, including reduction of depression, stress, anxiety and caregiver burden,[37 38] and have also been found to increase self-efficacy, self-esteem and strain of caregivers of adults with chronic conditions[39] as well as enhance the sense of competence, coping skills and strategies, and quality of life in caregivers of older adults.[40] A recently published systematic review on dyadic psychological eHealth interventions demonstrated that by integrating the dyadic approach with technology, interventions were found most suitable to meet caregivers' needs, as well as feasible and cost effective.[41] The current review aims to add more accurate knowledge on the subject by considering dyadic romantic partner relationships only, focusing on caregiver burden as the prespecified primary endpoint and conducting a meta-analysis limited to randomised controlled trials only.

## Aim of the study

To the best of our knowledge, no review or meta-analysis has systematically evaluated the current research on dyadic digital psychological interventions for informal caregivers. This study aims to systematically review the existing literature on the efficacy of dyadic digital psychological interventions designed for couples of informal caregivers and their care receivers within the illness context.

Specifically, we aim to examine:

1. The efficacy of dyadic digital psychological interventions for reducing caregivers' burden among informal caregivers who are in a romantic relationship with adults with a chronic illness.

2. The effectiveness of dyadic digital psychological interventions on secondary outcomes such as anxiety, depression, stress, quality of life, well-being, self-efficacy and relationship satisfaction among informal caregivers and care receivers.
3. The moderating effects of clinical and methodological factors on caregivers' burden.

## METHODS AND ANALYSIS

This protocol adopts the Preferred Items for Reporting of Systematic Reviews and Meta-Analyses Protocol (PRISMA-P) guidelines[42] (online supplemental appendix 1) and the Cochrane Risk of Bias tool to assess the risk of bias for randomised controlled trials.[43]

### Study registration

Following the PRISMA guidelines,[42] our systematic review protocol was registered with the International Prospective Register of Systematic Reviews (PROSPERO) on 3 March 2022 (registration number CRD42022299125). Any important protocol amendments will be recorded in PROSPERO and published with the results of the review.

### Eligibility criteria

Inclusion and exclusion criteria are categorised by population, interventions, comparators, outcomes and study design (PICOS).[44]

Eligible studies for inclusion will be in English language peer-reviewed studies or unpublished studies such as doctoral thesis to reduce publication bias.

### Population

We will include studies examining adults (18 years or older) who provide informal care and their care receivers in a romantic relationship. Thus, we will not include studies that include formal caregivers; studies in which the sample included caregiving parents, siblings or other non-romantic relationships; and finally, studies that do not collect data on both informal caregivers and care receivers. No restrictions have been placed on participants' gender and ethnicity.

### Interventions

Any psychological, psychoeducational, psychosocial or combination of psychological interventions using specific therapeutic principles and techniques delivered via a technological tool (eg, computer, telephone and videos). Any other type of interventions such as pharmacological interventions, acupuncture, smoking cessation, etc will be excluded. All the interventions included must be dyadic interventions addressing, targeting and providing data on both informal caregivers and care receivers. Individual intervention or studies that provide data on either caregiver or care receiver only will not be included in the current review.

### Comparator

Both active and inactive comparators will be eligible, such as no intervention, usual care, waiting list and attention control.

### Outcomes
#### Primary outcomes

Studies will be included only if one or more standardised measurements of caregivers' burden have been used.

#### Secondary outcomes

Secondary outcomes of interest are quantitative measurements listed below:
► Anxiety;
► Depression;
► Stress;
► Quality of life;
► Well-being;
► Self-efficacy.

### Study design

We will include only randomised controlled trials to optimise the internal validity of this review.

### Information sources
#### Search strategy

Two reviewers (MS and VB) will independently search the following database for relevant articles: PubMed, Embase, the Cochrane Library, Cinhal, Scopus, PsycINFO and MEDLINE.

For each database, a comprehensive search strategy will be developed under the PICO framework.[44] Boolean and truncation operators will be used to systematically combine search terms and to screen for a list of prespecified search terms in titles and abstracts (online supplemental appendix 2).

No restrictions will be placed on the publication period. The reference lists of all selected articles and relevant systematic review will be manually screened to identify further relevant records for possible inclusion.[45 46]

Additionally, studies from grey literature (eg, thesis dissertation and conference papers/abstract) will be searched in OpenGrey (https://opengrey.eu/) and will be included in case they fulfil the inclusion criteria.

Data will be exported to Endnote X8 (Clarivate Analytics, Philadelphia, Pennsylvania, USA) and Excel for the title and abstract screening.

### Study records
#### Data management

Data will be imported into Endnote X8 (Clarivate Analytics, Philadelphia, Pennsylvania, USA), and duplicate records will be removed, while (if possible) Stata V.16.0 (StataCorp) will be used for meta-analysis.

#### Selection process

Two reviewers will independently screen study titles and abstracts retrieved from searches; they will perform full paper checks of identified potentially eligible studies.

The PICOS criteria will be used to select eligible studies. In the results, manuscript will be documented and reported the overall reasons for exclusion at the full-text screening, using a PRISMA flow chart. A detailed overview of the reasons for the inclusion/exclusion of studies at the PICOS item level will be presented in a table in the results manuscript. Disagreements between reviewers will be solved by consulting a third reviewer. The overall selection process will be represented as a PRISMA flow diagram.

## Data extraction
All reviewers will be involved in reaching a consensus on the data to be extracted and the terminology used before the start of the screening. Data extraction will be conducted independently by two reviewers (MS and VB). One reviewer (MS) will screen the titles of each study based on the eligibility criteria, and when these are deemed relevant, she will go through abstracts. Once narrowed down by abstracts, a full-text review process will be completed in duplicate by two reviewers (MS and VB) for studies that met the eligibility criteria at screening and for studies with unclear relevance. If there are any disagreements, a third independent reviewer (either NV or RD) will be consulted to resolve discrepancies. Original authors of studies identified will be contacted if the full-text paper was not available, or the relevance of a paper was unclear. We will assess the inter-rater agreement by kappa statistic using GraphPad Software. A kappa value of 0.61–0.80 reflects substantial agreement, and a kappa value of 0.81–1.00 reflects (almost) perfect agreement.[47]

## Risk of bias
The Cochrane Collaboration's Risk for Bias tool will be used to evaluate the risk of bias and the quality of the studies in the form of randomised controlled trial.[43] Thanks to this tool each domain of potential bias will be classified as 'low risk', 'unclear risk' or 'high risk'.[48]

Two researchers will work independently. Any disagreement will be solved by a third reviewer.

## Data synthesis
A narrative synthesis will be undertaken to present and explain the findings. Two reviewers (MS and VB) will independently read the selected studies and will propose an initial framework for its synthesis. These frameworks will serve as a basis for a consensus synthesis framework, decided by all reviewers in a group discussion. All reviewers will be involved in writing the narrative synthesis, following a joint iterative process of organising and cross validating the reported results.

### Meta-analysis
If data allow, a meta-analysis will be performed to explore the effectiveness of dyadic digital psychological interventions for reducing caregivers' burden among informal caregivers by calculating post-treatment between-group standardised mean effect sizes for the primary outcome using Hedges' g. In addition, the effectiveness of dyadic digital psychological interventions will be examined on selected secondary outcomes (eg, depression, anxiety, stress, quality of life, well-being and self-efficacy) among both informal caregivers and/or care receivers.

Dichotomous outcomes will be measured by risk ratio and its 95% CI. Continuous outcomes will be measured by calculating the mean difference (MD) with 95% CI when the studies use the same instrument for assessing the outcome. The standardised MD will be used when studies use different instruments.

### Assessment of heterogeneity
A random-effect model will be performed for all the analyses because of the potential heterogeneity among clinical trial results. The heterogeneity between-study will be measured using the Cochrane's test of heterogeneity (Q) and will be reported using $I^2$ statistics, alongside CIs.[49 50] The indirect comparisons between technology will be assessed using the Bucher and Glenny method.[51]

### Moderator analyses
Prespecified subgroup analyses will be performed to assess the potential association of some clinical factors with the primary outcome. Subgroups were defined according to (1) study quality characteristics: control type (active vs non-active); (2) informal caregivers characteristics: age, women percentage and education; (3) patient characteristics: age, women percentage, education and type of illness; and (4) intervention characteristics: number of sessions, duration, type of therapy (eg, CBT vs mindfulness) and modality tool used to deliver the intervention (eg, telephone vs video).[52]

### Dealing with missing data
Intention-to-treat data will be used when possible.[53]

### Confidence in cumulative evidence
The Grading of Recommendations Assessment, Development and Evaluation (GRADE)[54 55] tool will be used to assess the certainty of the evidence of both primary outcome and secondary outcomes as 'high', 'moderate', 'low' or 'very low', according to the GRADE approach. A summary of evidence will be presented in a table.

### Patient and public involvement
There was no patient and public involvement in the development of this protocol.

## ETHICS AND DISSEMINATION
Any identifiable patient data will be excluded; thus, ethical approval and participant consent are not required. Findings will be disseminated through peer-reviewed publications, presentations at academic conferences and lay summaries for various stakeholders.

**Author affiliations**
[1]Department of Psychology, Bar-Ilan University, Ramat Gan, Israel
[2]Department of Psychology, Universita Cattolica del Sacro Cuore, Milano, Italy
[3]School of Social Work, Bar-Ilan University, Ramat Gan, Israel

[4]Department of Behavioural Science and Learning, Linkoping University, Linköping, Sweden
[5]Department of Clinical Neuroscience, Karolinska Institute, Stockholm, Sweden
[6]Psychology Research Laboratory, IRCCS Istituto Auxologico Italiano, Ospedale San Giuseppe, Oggebbio (VCO), Italy

**Contributors** MS contributed to the conception and design of the study and wrote the manuscript. NV contributed to the conception and design of the study and critically revised the manuscript drafts. VB, GA, RD and GP critically revised the study design, assisted in data synthesis and analysis, and revised the manuscript's final draft. All authors approved the final manuscript.

**Funding** This work is funded by the European Union's Horizon 2020 research and innovation program under the Marie-Sklodowska Curie grant agreement no 814072.

**Competing interests** None declared.

**Patient and public involvement** Patients and/or the public were not involved in the design, conduct, reporting or dissemination plans of this research.

**Patient consent for publication** Not applicable.

**Provenance and peer review** Not commissioned; externally peer reviewed.

**ORCID iDs**
Michelle Semonella http://orcid.org/0000-0003-0284-0575
Vanessa Bertuzzi http://orcid.org/0000-0002-8541-5357
Noa Vilchinsky http://orcid.org/0000-0003-4965-4745

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
