## [Reviewer comments · BMJ Open]

ARTICLE DETAILS

TITLE (PROVISIONAL)	Applying dyadic digital psychological interventions for reducing caregiver burden in the illness context: A systematic review and a meta-analysis protocol
AUTHORS	Semonella, Michelle; Bertuzzi, Vanessa; Dekel, Rachel; Andersson, Gerhard; Pietrabissa, Giada; Vilchinsky, Noa

VERSION 1 – REVIEW

REVIEWER	Lucassen, Peter Radboud University Medical Centre, Department of Primary and Community Care
REVIEW RETURNED	27-Dec-2022

GENERAL COMMENTS	This is a protocol for a systematic review of the effectiveness of a digital dyadic psychological intervention for caregiver burden. The subject is important and novel, because it is directed at the dyad and not at the individual. Major comments 1. The databases for the planned systematic review do not include Embase (the difference between Scopus and Embase is not clear), the Cochrane Library and Cinahl (which includes a lot of papers about caregiving). Because of the lack of research in this field (as the authors state in the Introduction) it is important to include these databases as well.2. Page 11, line 55-58: the external validity is not determined by design but by selection of participants. The choice for RCTs is OK concerning internal validity for studies determining effectiveness, provided that the study has a low risk for bias. Minor comments 1. In the Abstract the authors write: 'However, there is a lack of research examining the effectiveness of digital dyadic solutions.' And they continue with: 'Thus, the review aims to examine the existing literature on the efficacy....' This needs rephrasing as there is a bit contradiction in it (page 2 line 19-26).2. The references (11-15) are a bit old. Suggestions for more recent references from the field of palliative care: (1) Choi S, Seo JY. Analysis of caregiver burden in palliative care: an integrated review. Nursing Forum 2019;54:280-290. (2) Pop RS, Puia A, Mosoiu D. Factors influencing the quality of life of the primary caregiver of a palliative patient: narrative review. J Pall Med 2022;25:813-829.3. On page 7 line 33-36, the authors state that dyadic interventions have been shown to be successful, but most of the references are not RCTs.
---

	4. Search strategy: I would prefer a slightly more elaborate description of the strategy, for example: we had a search strategy consisting of the following strings (a then name these strings) connected with Boolean operators.... (page 12). 5. There is no mentioning of handling duplicate publications (page 13). 6. Did the authors deliberately decide not to calculate the kappa statistic? This would be a method to determine how well the selection process has been. 7. Page 13, line 30-33: The selection process of the relevant studies will be...: this sentence does not belong here. The paragraph is about data extraction. The authors do not describe if data extraction will be performed independently by two researchers.
--	--

REVIEWER	Atherton, John Royal Brisbane and Women's Hospital and University of Queensland, Department of Cardiology
REVIEW RETURNED	21-Jan-2023

GENERAL COMMENTS	This manuscript describes the protocol for a systematic review and meta-analysis of randomised controlled trials evaluating dyadic digital psychological interventions for reducing caregiver burden in the illness context. The manuscript is well written, and the approach outlined is appropriate. The protocol is following the PRISMA guidelines and has been registered with PROSPERO. The eligibility criteria will be formulated according to the PICOS framework. The Cochrane Collaboration's Risk for Bias tool will be used to assess risk of bias. The strength of evidence will be evaluated using the GRADE tool. This is an important topic, especially given the aging population and increasing prevalence of chronic disease, thereby increasing the need for caregiver support as an integral component of healthcare. This will inform the development, further evaluation, and implementation of dyadic digital psychological interventions in the future. Minor comments:  1. The authors should mention the recently published systematic review of dyadic psychosocial eHealth interventions (Shaffer KM et al. J Med Internet Res 2020;22:e15509), and how their more focussed systematic review differs from Shaffer et al. (e.g. only considering dyadic romantic partner relationships; limited to randomised controlled trials; caregiver burden as the prespecified primary endpoint). 2. The authors may also consider evaluating the moderating effect of cognition and frailty for both the caregivers and care receivers (if evaluated) on outcomes.
--

VERSION 1 – AUTHOR RESPONSE

Reviewer: 1

Major comments

C1: "The databases for the planned systematic review do not include Embase (the difference between Scopus and Embase is not clear), the Cochrane Library and Cinahl (which includes a lot of papers about caregiving). Because of the lack of research in this field (as the authors state in the Introduction) it is important to include these databases as well".

A1: "The authors would like to thank Reviewer 1 for this important comment. The requested databases were included. Please see page 2, line 12, and page 9, line 2".

C2: "Page 11, line 55-58: the external validity is not determined by design but by selection of participants. The choice for RCTs is OK concerning internal validity for studies determining effectiveness, provided that the study has a low risk for bias".

A2: "We thank the reviewer for pointing this out. We reworded the sentence accordingly. Please see page 8, line 25".

Minor comments

C1: "In the Abstract the authors write: 'However, there is a lack of research examining the effectiveness of digital dyadic solutions.' And they continue with: 'Thus, the review aims to examine the existing literature on the efficacy....' This needs rephrasing as there is a bit contradiction in it (page 2 line 19-26)".

A: "Thank you for your comment. We have revised the text to address your concern and we hope that it is now clearer and not contradictive. Please see page 2, lines 7-8".

C2: "The references (11-15) are a bit old. Suggestions for more recent references from the field of palliative care: (1) Choi S, Seo JY. Analysis of caregiver burden in palliative care: an integrated review. *Nursing Forum* 2019;54:280-290. (2) Pop RS, Puia A, Mosoiu D. Factors influencing the quality of life of the primary caregiver of a palliative patient: narrative review. *J Pall Med* 2022;25:813-829."

A2: "The authors would like to thank the reviewer for this valuable suggestion. These references have been updated. Please see page 4, line 17".

C3: "On page 7 line 33-36, the authors state that dyadic interventions have been shown to be successful, but most of the references are not RCTs".

A3: "Thank you for this comment. The references have been updated, taking into account only RCTs. Please see page 5, line 18-19".

C4: "Search strategy: I would prefer a slightly more elaborate description of the strategy, for example: we had a search strategy consisting of the following strings (a then name these strings) connected with Boolean operators.... (page 12)".

A4: "Following your comment, the full search strategy has been included as a supplementary file".

C5: "There is no mentioning of handling duplicate publications (page 13)".

A7: "Thank you for your comment. We added this information. Please see page 9, lines 19".

C6: "Did the authors deliberately decide not to calculate the kappa statistic? This would be a method to determine how well the selection process has been."

A6: "Thank you for this comment. We now include kappa statistic. Please see page 10, lines 7-9".

C7: "Page 13, line 30-33: The selection process of the relevant studies will be...: this sentence does not belong here. The paragraph is about data extraction. The authors do not describe if data extraction will be performed independently by two researchers."

A7: "We wish to thank the reviewer for this important comment. Indeed, data extraction will be performed independently by two researchers. We declared this info now, please see page 9, 33. Moreover, we deleted the sentence that you mentioned as not belonging to this paragraph, see on page 9, lines 32-33".

Reviewer: 2

Minor comments:

C1: "The authors should mention the recently published systematic review of dyadic psychosocial eHealth interventions (Shaffer KM et al. J Med Internet Res 2020;22:e15509), and how their more focussed systematic review differs from Shaffer et al. (e.g. only considering dyadic romantic partner relationships; limited to randomised controlled trials; caregiver burden as the prespecified primary endpoint)".

A1 "Thank you for the suggestion, this is a very interesting paper. Following your advice, we insert this systematic review in the introduction paragraph. Also, we focused on the differences between this systematic review and our systematic review. Please see page 6, lines 9-14.

C2: "The authors may also consider evaluating the moderating effect of cognition and frailty for both the caregivers and care receivers (if evaluated) on outcomes".

A2: "We appreciate the reviewer's insightful suggestion and agree that it would be useful to demonstrate that. However, since we will focus on digital dyadic solutions, such intervention trials usually consider cognitive impairments and frailty as exclusion criteria to take part in the interventions. Thus, we most probably are unable to discern these important aspects".

VERSION 2 – REVIEW

REVIEWER	Atherton, John Royal Brisbane and Women's Hospital and University of Queensland, Department of Cardiology
REVIEW RETURNED	31-Mar-2023

GENERAL COMMENTS	The authors have adequately addressed my comments. Minor typo in new text on page 13 (left line margin 6): Change "and duplicate record will be removed" to "and duplicate records will be removed". The manuscript reads well. I wish them well with their study.
---